# Characterization of geographic mobility among participants in facility- and community-based tuberculosis case finding in urban Uganda

Katherine O. Robsky[1,2]*, David Isooba[2], Olga Nakasolya[2], James Mukiibi[2], Annet Nalutaaya[2], Peter J. Kitonsa[2], Caleb Kamoga[2], Yeonsoo Baik[1,2], Emily A. Kendall[2,3], Achilles Katamba[2,4], David W. Dowdy[1,2,3]

1 Department of Epidemiology, Johns Hopkins Bloomberg School of Public Health, Baltimore, Maryland, United States of America, 2 Uganda Tuberculosis Implementation Research Consortium, Makerere University, Kampala, Uganda, 3 Johns Hopkins School of Medicine, Baltimore, Maryland, United States of America, 4 Department of Medicine, Clinical Epidemiology and Biostatistics Unit, College of Health Sciences, Makerere University, Kampala, Uganda

* krobsky1@jhmi.edu

**Data Availability Statement:** The dataset used for this analysis is available on the Johns Hopkins

## Abstract

### Background

International and internal migration are recognized risk factors for tuberculosis (TB). Geographic mobility, including travel for work, education, or personal reasons, may also play a role in TB transmission, but this relationship is poorly defined. We aimed to define geographic mobility among participants in facility- and community-based TB case finding in Kampala, Uganda, and to assess associations between mobility, access to care, and TB disease.

### Methods

We included consecutive individuals age ≥15 years diagnosed with TB disease through either routine health facility practices or community-based case finding (consisting of door-to-door testing, venue-based screening, and contact investigation). Each case was matched with one (for community-based enrollment) or two (health facility enrollment) TB-negative controls. We conducted a latent class analysis (LCA) of eight self-reported characteristics to identify and define mobility; we selected the best-fit model using Bayesian Information Criterion. We assessed associations between mobility and TB case status using multivariable conditional logistic regression.

### Results

We enrolled 267 cases and 432 controls. Cases were more likely than controls to have been born in Kampala (p<0.001); there was no difference between cases and controls for remaining mobility characteristics. We selected a two-class LCA model; the "mobile" class was perfectly correlated with a single variable: travel (>3 km) from residence ≥2 times per month.

University Data Archive: https://archive.data.jhu.edu/dataverse/stomp-tb.

**Funding:** This work was supported by the National Institutes of Health [R01HL138728 to D.W.D. and K08AI127908 to E.A.K.] and the Fogarty-Fulbright Fellowship in Public Health [FICD43TW010540 to K.O.R.].

**Competing interests:** The authors have declared that no competing interests exist.

Mobility was associated with a 28% reduction in odds of being a TB case (adjusted matched odds ratio 0.72 [95% confidence interval 0.49, 1.06]).

## Conclusion

Frequency of out-of-neighborhood travel is an easily measured variable that correlates closely with predicted mobility class membership. Mobility was associated with decreased risk of TB disease; this may be in part due to the higher socioeconomic status of mobile individuals in this population. However, more research is needed to improve assessment of mobility and understand how mobility affects disease risk and transmission.

## Introduction

Tuberculosis (TB) is a leading cause of morbidity and mortality globally, causing an estimated 10.0 million cases and 1.2 million deaths in 2018 [1,2]. Mobile and migratory individuals are at increased risk for TB infection and disease [3]. Mobile individuals may be more likely to acquire or transmit TB [4,5] and often experience barriers to TB diagnosis and treatment [6,7]. While most research on TB and migration is focused on international migration [3], individuals with high internal mobility such as rural-urban migration [8,9], labor migration [10], and nomadic populations [11–13] have also been shown to be at high risk for TB and to experience barriers to TB care.

Many studies on the association between mobility and infectious disease have focused on HIV in defined populations that are known to experience high mobility. For example, studies have shown truck drivers to be at high risk of acquiring [14–16] and transmitting HIV [17] and to have limited access to health care [18]. Agricultural migrant workers have also been shown to be at increased risk for HIV [19,20]. However, mobility may arise from a variety of experiences including marriage, work, and education [21], and research into relationships between mobility and disease lacks a standard measure of geographic mobility. Studies of mobility and HIV have considered frequency [20,22–24] and duration of travel from home [22], number of nights spent away from home [22–25], circular or temporary migration [26], as well as distance traveled or internal borders crossed [21,22,24], but there has been little research investigating such measures among populations at risk for TB. The extent to which mobility measures applied to individuals at risk for HIV are relevant to TB, a disease that shares many risk factors but has a different mechanism of transmission [27], is not known.

We aimed to develop a broader understanding of mobility patterns in relation to TB. We described mobility patterns among participants in a facility- and community-based TB case finding study in Kampala, Uganda. We also assessed relationships between mobility and TB risk in this setting.

## Materials and methods

### Study population

This analysis was conducted as part of the STOMP-TB (Strategies for Treating, Observing, Managing, and Preventing Tuberculosis) study, an ongoing population-based study in a densely populated area consisting of 37 contiguous administrative zones in Kampala, Uganda (estimated population: 50,436, total area 2.2 km$^2$ [28]). The STOMP-TB study enrolls patients through two mechanisms: health facility TB diagnostic testing (from May 2018-July 2021) and

active community outreach and testing (Phase 1 conducted from February to December 2019).

For this analysis, we enrolled all consenting patients age ≥15 years who presented for TB testing at one public and three private outpatient TB Diagnosis and Treatment Units in the study area from February 2019 (when we began collecting information on mobility) to August 2020 (at the administrative stopping date for this analysis). Residence within the study area was required for enrollment prior to January 2020 at Kisugu Health Center (the public facility) and at other facilities throughout the study period; beginning January 2020, we also enrolled participants from Kisugu Health Center regardless of their residence. We identified TB cases as patients diagnosed with pulmonary TB by the treating clinician, regardless of microbiological test result; however, most cases were confirmed with sputum Xpert MTB/RIF or Xpert Ultra (Cepheid, Inc., Sunnyvale, CA, USA). For each case, two controls matched by facility, approximate date of enrollment, and location of residence (within study area vs. outside study area, for the participants from Kisugu Health Center) were enrolled. Health facility controls were randomly selected from eligible individuals who presented to the same treating facility and were tested for pulmonary TB but had a negative Xpert result and were not empirically treated for TB.

Additionally, from February through November 2020, we identified TB cases in the community through a coordinated campaign of active case finding activities including door-to-door testing, venue-based screening events, and contact tracing, using Xpert Ultra. Individuals with positive Xpert Ultra sputum results were enrolled as community cases, and each case was matched with one community control who resided in the same zone but had a negative Xpert Ultra result. Data collection for all participants included interviews and abstraction from clinical and laboratory records.

## Measurement of components of geographic mobility

We defined *a priori* eight components of mobility (Table 1) and collected self-reported information through participant interviews using a tool developed for and pilot tested in this study population (S1 Appendix). We dichotomized average number of nights spent away from primary residence per month at ≥10 days per month based on a natural break in the data; in a sensitivity analysis we also considered ≥1 day per month. Both definitions align with those found in the literature [22,23]. We defined travel outside of the participant's neighborhood as travel >3 km from their primary residence; this represents leaving the study area, likely use of a car, bus, or taxi, and the potential for TB exposure outside the immediate vicinity of the participant's local community. Research assistants provided landmarks and Google maps to help participants identify locations they may have visited >3 km from home. We used the overall median in the population to define the cutoff for frequency and duration of travel >3 km from home; we also considered the 75[th] quartile as a cutoff and modeling each of the variables as Poisson-distributed (non-dichotomized) as sensitivity analyses. The remaining measures and their categorization are described in Table 1.

## Classification of mobility

We used latent class analysis (LCA) to identify patterns of data that could inform our definition of mobility. The construct of mobility as a latent variable that influences eight observable indicator variables is shown in Fig 1. We conducted a LCA using structural equation models with a logit link using these eight variables as defined in Table 1 for the entire study population [29]. To determine the number of classes that provided the best model fit, for each LCA we considered models with between one and four classes and selected the model with the lowest

**Table 1. Components of geographic mobility.**

| Variable | Mobile | Non-mobile |
|---|---|---|
| Place of Birth | outside Kampala, including outside Uganda | Kampala |
| Time lived within 3 km of your current residence* | < 1 year | ≥1 year |
| Have another residence | yes | no |
| Number of nights per month spent not at primary residence* | ≥10 night(s) per month | <10 nights per month |
| *Sensitivity analysis 3* | *≥1 night(s) per month* | *0 nights per month* |
| Frequency of visiting a taxi park** | ≥1 time per month | 0 times per month |
| Frequency of travel >3 km from residence* | ≥2 times per month (median) | <2 times per month |
| *Sensitivity analysis 1* | *Poisson-distributed variable (no dichotomization)* | |
| *Sensitivity analysis 3* | *≥8 times per month (75th percentile)* | *<8 times per month* |
| Time spent during travel >3 km away from residence* | ≥3 hours per trip (median) | <3 hours per trip |
| *Sensitivity analysis 2* | *Poisson-distributed variable (no dichotomization)* | |
| *Sensitivity analysis 5* | *≥8 hours per trip (75th percentile)* | *<8 hours per trip* |
| Number of times traveled outside of Kampala in the past 12 months | ≥1 trip in the last year | 0 trips in the last year |

*For individuals with multiple places of current residence, the one within the study area was selected as the primary residence. For participants from outside the study area (enrolled at Kisugu Health Center), the residence where the participant lives most of the time was selected as the current primary residence.

**Any of several individually-queried taxi parks near the study area (Kisugu Taxi stage, Namuwongo Taxi Stage, Old Taxi Park, New Taxi Park, or Ssali Stage Wabigalo). Note that in Uganda, a taxi refers to a van or minibus that frequently picks up and drops of passengers along a specific route, akin to a public bus route in other settings.

Bayesian Information Criterion (BIC). We characterized the classifications of mobility based on the marginal means estimated by LCA, and we assigned each participant to their most probable mobility class. We then identified the characteristics most common among the "mobile" latent class and developed a calculated definition of mobility using observed, rather than latent, variables that most closely aligned with the classes predicted by the LCA. The calculated, or observed, classes of mobility were used for further analyses. We also conducted two sensitivity analyses by repeating the LCA process stratified by 1) case status (case vs. control) and 2) enrollment mechanism (health facility vs. community).

Using our calculated definition of mobility, we used log binomial regression to estimate the unadjusted prevalence ratios for the association of mobility and demographic, socioeconomic, and TB risk factors, stratified by TB case status. We then assessed the association of mobility with TB case status (case vs. control) using conditional logistic regression on case-control pairs/triads matched for enrollment method (facility vs. community), location of residence (within vs. outside study area for health facility enrollment, zone of residence for community enrollment), and facility (for health facility enrollment only) adjusting for possible confounders. All potential confounders were identified a priori as characteristics believed to be associated with mobility and previously shown to be associated with TB disease, including demographic, socioeconomic, and clinical and behavioral risk factors, and were included in the final model regardless of statistical significance. All analyses were conducted using Stata version 16, using the 'gsem' package for latent class analysis.

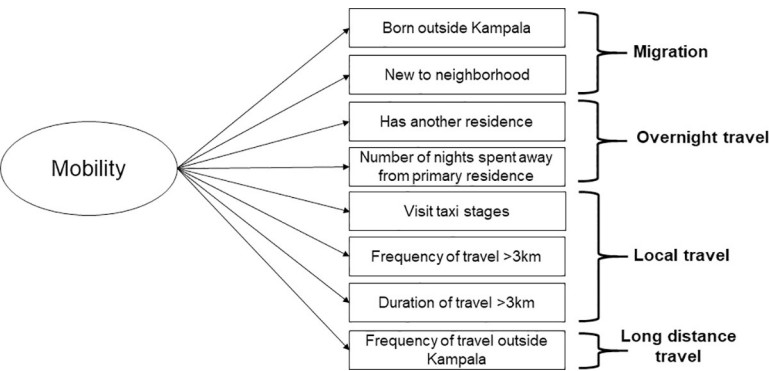

**Fig 1. Construct of geographic mobility as a latent variable.** Geographic mobility is conceptualized as a latent (unobserved) variable which influences eight observable indicator variables that capture migration, overnight travel, local travel, and long distance travel. Association of mobility with TB status.

## Ethical considerations

The study was approved by the Johns Hopkins Bloomberg School of Public Health Institutional Review Board (IRB Number 11353) and the Higher Degrees, Research and Ethics Committee of the Makerere University School of Public Health, Kampala-Uganda (Study Protocol Number 544). All participants provided written informed consent (or written assent and parental consent for those 15–17 years old) for all study activities.

## Results

### Study population

We enrolled a total of 699 participants: 499 from health facilities (167 TB cases and 332 matched controls; two cases had only one control enrolled) and 200 from community case finding activities (100 TB cases and 100 matched controls). Overall, 328 (47%) were female and the median age was 31 years (IQR 23–41). The majority of participants were born outside of Kampala (84%, n = 584) and had traveled outside of Kampala at least once in the last year (76%, n = 531) (Table 2). Other indicators of mobility were reported less frequently: 19% (n = 134) had moved to the area within the last year, 15% (n = 106) had a second residence, 10% (n = 70) spent more than ten nights per month away from their primary residence, and 27% (n = 187) visited a taxi stage at least once a week. Participants reported a median of two trips (IQR 0,8) more than 3 km from home (representing leaving the neighborhood) per month and spending a median of three hours (IQR 0,8) away from home during such trips.

Individuals with TB generally had similar mobility characteristics compared to TB-negative controls for both health facility and community enrolled participants, although they were more likely to have been born in Kampala (23% vs. 12%, p<0.001). Community-enrolled controls were the most likely to report having a second residence (22%), spending more than 10 nights per month away from their primary residence (17%), and visiting a taxi stage at least once per week (60%).

### Classification of mobility

The model with two latent classes had the best fit based on BIC (S1 Table). One class was characterized as "mobile" based on higher estimated marginal means for the following characteristics: Spending ≥10 nights away from their primary residence each month (14% vs 6%), visiting a taxi stage at least once a week (36% vs. 18%), traveling more than 3 km from

**Table 2. Characteristics of people with and without TB in an urban Ugandan community.**

| | Total N = 699 | Health Facility Enrollment | | Community Case Finding Enrollment | |
|---|---|---|---|---|---|
| | | TB Case N = 167 | Control N = 332 | TB Case N = 100 | Control N = 100 |
| | | N (%) | N (%) | N (%) | N (%) |
| **Mobility Characteristics** | | | | | |
| Born outside Kampala | 583 (84%) | 124 (75%) | 289 (87%) | 81 (81%) | 89 (89%) |
| Lived in neighborhood <1 year | 143 (19%) | 29 (18%) | 66 (20%) | 20 (20%) | 19 (19%) |
| Have another residence | 106 (15%) | 16 (10%) | 52 (16%) | 16 (16%) | 22 (22%) |
| Spends ≥10 nights away from primary residence | 70 (10%) | 10 (6%) | 33 (10%) | 10 (10%) | 17 (17%) |
| Visited taxi stage ≥1 time per week | 187 (27%) | 29 (17%) | 58 (18%) | 40 (40%) | 60 (60%) |
| Travel 3km ≥2 times per month (median, IQR) | 2 (0,8) | 1 (0,10) | 2 (0,6) | 3 (0, 11.5) | 3 (0,20) |
| Spend ≥3 hours away when traveling 3km (median, IQR) | 3 (0,8) | 3 (0,7) | 3 (0,8) | 4 (0,8) | 2.5 (0,8) |
| Ever traveled outside Kampala in last year | 531 (76%) | 121 (73%) | 252 (76%) | 77 (77%) | 81 (81%) |
| **Demographic Characteristics** | | | | | |
| Age in years | | | | | |
| 15–24 | 197 (28%) | 39 (23%) | 91 (27%) | 26 (26%) | 41 (41%) |
| 25–34 | 213 (31%) | 51 (31%) | 91 (27%) | 40 (40%) | 31 (31%) |
| 35–44 | 166 (24%) | 46 (28%) | 80 (24%) | 20 (20%) | 20 (20%) |
| ≥45 | 122 (18%) | 30 (18%) | 70 (21%) | 14 (14%) | 8 (8%) |
| Male sex | 370 (53%) | 112 (68%) | 163 (49%) | 58 (58%) | 37 37%) |
| **Socioeconomic Status** | | | | | |
| Highest completed education | | | | | |
| None | 416 (60%) | 59 (35%) | 127 (38%) | 31 (31%) | 25 (25%) |
| Certificate | 242 (35%) | 95 (57%) | 187 (56%) | 62 (62%) | 72 (72%) |
| Degree/further studies | 40 (6%) | 12 (7%) | 18 (5%) | 7 (7%) | 3 (3%) |
| Able to read/write without difficulty | 238 (34%) | 62 (37%) | 124 (37%) | 34 (34%) | 18 (18%) |
| Occupation | | | | | |
| Employed | 517 (74%) | 126 (76%) | 238 (72%) | 84 (84%) | 69 (69%) |
| Unemployed | 76 (11) | 27 (16%) | 40 (12%) | 6 (6%) | 3 (3%) |
| Student or housewife | 105 (15%) | 13 (8%) | 53 (16%) | 10 (10%) | 28 (28%) |
| Household income quartile | | | | | |
| 1st (lowest) | 203 (29%) | 55 (33%) | 89 (27%) | 28 (28%) | 31 (31%) |
| 2nd | 149 (21%) | 35 (21%) | 79 (24%) | 21 (21%) | 14 (14%) |
| 3rd | 198 (28%) | 48 (29%) | 91 (27%) | 25 (25%) | 34 (34%) |
| 4th (highest) | 149 (21%) | 29 (17%) | 73 (22%) | 26 (26%) | 21 (21%) |
| **Risk factors for TB** | | | | | |
| HIV Positive | 164 (24%) | 52 (31%) | 92 (28%) | 12 (12%) | 8 (8%) |
| Previous TB Treatment | 77 (11%) | 36 (22%) | 34 (10%) | 6 (6%) | 1 (1%) |
| Limitations in any of the EQ-5D domains | 407 (58%) | 124 (75%) | 188 (57%) | 50 (50%) | 45 (45%) |
| Ever had household TB contact | 155 (22%) | 42 (25%) | 65 (20%) | 31 (31%) | 17 (17%) |
| Known a TB case (past 12 months) | 182 (26%) | 50 (30%) | 76 (23%) | 33 (33%) | 23 (23%) |
| Household has ≥3 people | 322 (46%) | 67 (40%) | 164 (49%) | 43 (43%) | 48 (48%) |

residence ≥2 times per month (100% vs. 4%), and spending ≥3 hours away from home when traveling more than 3 km from residence (87% vs 15%) (Fig 2, S2 Table). Sensitivity analyses stratifying the LCA model by case status or enrollment method yielded similar results (S3 & S4 Tables). These results did not change in sensitivity analyses in which we considered different measures for the number of nights spent away from their primary residence and the frequency and duration of travel >3 km (S5 & S6 Tables).

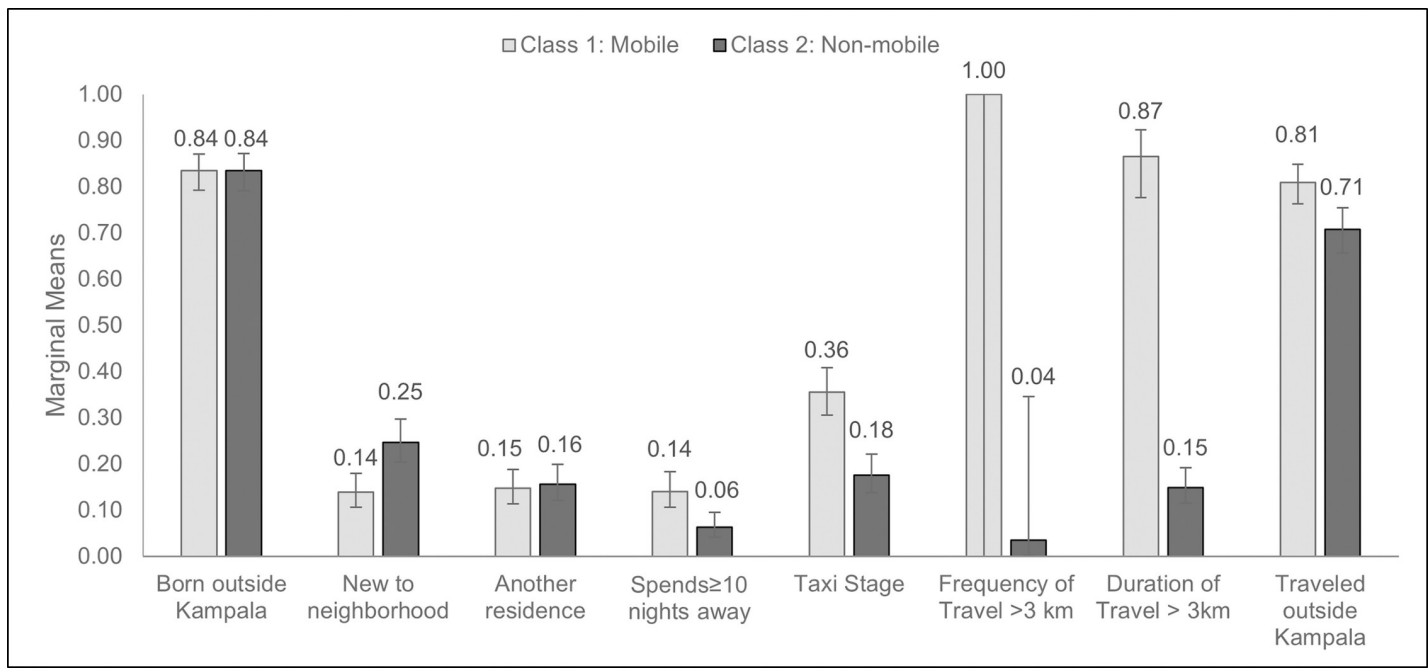

**Fig 2. Marginal means for latent classes of mobility.** The estimated mean for each observed item (of the eight observed varables) is shown for two latent classes: mobile and non-mobile. These values are also presented in S2 Table.

We assigned the most probable class as predicted by the LCA model to each individual in our entire participant population, resulting in 369 (53%) participants classified as mobile: 78 (47%) health facility TB cases, 172 (52%) health facility controls, 57 (57%) community TB cases, 62 (62%) community controls. There was perfect correlation between being assigned to the "mobile" class and traveling >3 km from home residence ≥2 times per month; we used this measure as a proxy for mobility in subsequent analyses.

### Characteristics associated with mobility

Among TB cases, mobile individuals (defined as traveling >3 km from home residence ≥2 times per month) were more likely to be male (PR 1.29, 95% CI 0.99, 1.69) and to be in the highest income quartile (PR 1.28, 95% CI 0.93, 1.7) compared to non-mobile individuals; they were also less likely to be unemployed (PR 0.21, 95% CI 0.08, 0.52) and more likely to have completed a degree (PR 1.41, 95% CI 0.89, 2.22) (Table 3). Among controls, mobile individuals were similarly more likely to be male (PR 1.53, 95% CI 1.28, 1.82) and to be in the highest income quartile (PR 1.46, 95% CI 1.15, 1.84), and less likely to be unemployed (PR 0.51, 95% CI 0.33, 0.79). Unlike among cases, there was no association between mobility and degree completion among controls (PR 0.91, 95% CI 0.54, 1.52).

Among individuals with TB who reported any TB symptoms, the median duration of symptoms was 8 weeks among those enrolled at the health facility and 4 weeks among those enrolled during community case finding; there was no difference when stratified by mobile classification (Wilcoxon rank-sum p-value 0.66 [health facility] and 0.30 [community]) (S7 Table).

### Association of mobility with TB status

Among TB cases, 51% (n = 135) were classified as mobile using our proxy measure (defined as traveling >3 km from home residence ≥2 times per month), compared to 54% (n = 234) of

**Table 3. Association of demographic, socioeconomic, and TB risk characteristics with mobility*.**

| | TB Cases | Controls |
|---|---|---|
| | Unadjusted Prevalence Ratio (95% CI) | Unadjusted Prevalence Ratio (95% CI) |
| **Enrollment method** | | |
| Health Facility | *Reference* | *Reference* |
| Community | 1.22 (0.96, 1.54) | 1.20 (0.99, 1.44) |
| **Demographic factors** | | |
| Age in years | | |
| 15–24 | *Reference* | *Reference* |
| 25–34 | 1.35 (0.95, 1.92) | 1.27 (1.01, 1.61) |
| 35–44 | 1.25 (0.85, 1.83) | 1.21 (0.95, 1.56) |
| $\geq$45 | 1.53 (1.05, 2.24) | 1.15 (0.87, 1.51) |
| Male Sex | 1.29 (0.99, 1.69) | 1.53 (1.28, 1.82) |
| **Socioeconomic Status** | | |
| Highest completed education | | |
| None | *Reference* | *Reference* |
| Certificate | 1.35 (1.01, 1.79) | 1.25 (1.03, 1.52) |
| Degree/further studies | 1.41 (0.89, 2.22) | 0.91 (0.54, 1.52) |
| Able to read/write without difficulty | 0.83 (0.64, 1.08) | 0.80 (0.66, 0.98) |
| Occupation | | |
| Employed | *Reference* | *Reference* |
| Unemployed | 0.21 (0.08, 0.52) | 0.51 (0.33, 0.79) |
| Student or housewife | 0.52 (0.28, 0.97) | 0.46 (0.32, 0.65) |
| Household income quartile | | |
| $1^{st}$ (lowest) | *Reference* | *Reference* |
| $2^{nd}$ | 0.99 (0.69, 1.42) | 1.00 (0.75, 1.32) |
| $3^{rd}$ | 1.08 (0.78, 1.49) | 1.15 (0.89, 1.47) |
| $4^{th}$ (highest) | 1.28 (0.93, 1.7) | 1.46 (1.15, 1.84) |
| **TB Risk Factors** | | |
| HIV Positive | 0.79 (0.59, 1.09) | 0.81 (0.64, 1.02) |
| Previous TB Treatment | 0.82 (0.57, 1.19) | 0.89 (0.62, 1.26) |
| Limitation in any of the EQ-5D domains | 0.87 (0.69, 1.11) | 0.87 (0.73, 1.03) |
| Ever had household TB contact | 0.86 (0.65, 1.14) | 0.93 (0.74, 1.18) |
| Known a TB case (past 12 months) | 1.00 (0.77, 1.29) | 1.16 (0.96, 1.40) |
| Household has $\geq$3 people | 1.01 (0.79, 1.28) | 1.02 (0.86, 1.21) |

*defined as traveling >3km $\geq$2 times per month.

controls (p = 0.35). Mobility was associated with a 28% reduction in the odds of TB disease (adjusted matched odds ratio [aOR] 0.72, 95% CI 0.49, 1.06) (Table 4). Independent risk factors for TB included male sex, previous treatment for TB, and reporting any limitations in EQ-5D domains.

## Discussion

Mobility is a key factor in infectious disease epidemics, as it can propagate transmission of disease and create challenges in accessing health care. In this analysis of nearly 700 individuals tested for TB in urban Uganda, we found that mobility was best defined by the frequency of

**Table 4. Association of patient characteristics with TB case status.**

| | Unadjusted Matched Odds Ratio (95% CI) | Adjusted Matched Odds Ratio (95%CI) |
|---|---|---|
| Mobility (traveling >3km ≥2 times per month) | 0.81 (0.58, 1.11) | 0.72 (0.49, 1.06) |
| **Demographic characteristics** | | |
| Age in years | | |
| 15–24 | *Reference* | *Reference* |
| 25–34 | 1.62 (1.04, 2.51) | 1.16 (0.68, 1.97) |
| 35–44 | 1.51 (0.95, 2.39) | 0.97 (0.56, 1.68) |
| ≥45 | 1.33 (0.81, 2.18) | 0.73 (0.40, 1.32) |
| Male Sex | 2.12 (1.55, 2.90) | 2.06 (1.41, 3.02) |
| **Socioeconomic characteristics** | | |
| Highest completed education | | |
| None | *Reference* | *Reference* |
| Certificate | 0.91 (0.68, 1.34) | 1.66 (0.73, 1.86) |
| Degree/further studies | 1.47 (0.75, 2.88) | 1.66 (0.72, 3.83) |
| Able to read/write without difficulty | 1.26 (0.89, 1.77) | 1.26 (0.78, 2.02) |
| Occupation | | |
| Employed | *Reference* | *Reference* |
| Unemployed | 1.36 (0.80, 2.31) | 1.53 (0.82, 2.88) |
| Student or housewife | 0.37 (0.22, 0.63) | 0.54 (0.29, 1.01) |
| Income quartile | | |
| 1st (lowest) | *Reference* | *Reference* |
| 2nd | 0.90 (0.59, 1.39) | 0.85 (0.52, 1.39) |
| 3rd | 0.84 (0.57, 1.25) | 0.85 (0.53, 1.34) |
| 4th (highest) | 0.82 (0.52, 1.29) | 0.84 (0.49, 1.44) |
| **TB Risk Factors** | | |
| HIV Positive | 1.25 (0.84, 1.85) | 1.05 (0.66, 1.69) |
| Previous TB Treatment | 2.40 (1.49, 3.87) | 1.88 (1.11, 3.18) |
| Limitation in any of the EQ-5D domains | 1.86 (1.33, 2.62) | 1.93 (1.31, 2.83) |
| Ever had household TB contact | 1.62 (1.12, 2.34) | 1.41 (0.92, 2.16) |
| Known a TB case (past 12 months) | 1.52 (1.07, 2.17) | 1.42 (0.94, 2.14) |
| Household has ≥3 people | 0.71 (0.51, 0.98) | 0.85 (0.58, 1.25) |

trips greater than 3 km from home residence. Our results suggest that mobility may be associated with a reduction in TB risk, which may be explained by the higher socioeconomic status of mobile individuals in this population.

There is no consistently applied definition of mobility, but frequency of travel outside of the neighborhood has been used in other studies of mobility in sub-Saharan Africa [20,22–24]. A strength of our definition is that it was determined using a data-driven approach based on our latent class analysis. In our population, a single observed characteristic was found to perfectly predict membership in the mobile latent class. Using an observed rather than latent variable avoids challenges in the estimation of standard errors [30], enables straightforward replication in other studies, and may enhance transportability of the mobility definition. Some studies distinguish between internal migration and travel as components of mobility [21]; we considered both in our LCA but ultimately only used travel in our final definition. Additionally, our use of a 3 km cutoff, designed to capture travel beyond the participants'

neighborhood, has not been used in other studies of mobility. We did consider longer travel distance (outside of Kampala), which is similar to inter-district travel used in other studies [21], but found that shorter distance better distinguished the classes of mobility. We found that nights spent away from the home, a common component of mobility in other studies [20,22–25], was higher among the mobile group but was not a distinguishing characteristic in defining mobility. While the 3 km cutoff was chosen based on the context of the study setting, including population density and available modes of transportation, future studies should consider the use of a variable collecting a setting-appropriate measure of extra-neighborhood travel at least twice per month as an easily measured marker of mobility.

While we hypothesized that mobile populations would be at increased risk for TB disease (due to increased range of contacts) and may experience barriers to care (due to lack of consistent access to the same nearby facility), we instead observed a reduction in the odds of TB disease among mobile individuals compared to non-mobile individuals. The effect of mobility on disease outcomes is likely driven by the cause and context of mobility, which we were unable to assess in this analysis. There is conflicting evidence as to whether mobile populations have more or less exposure to education campaigns and whether they experience greater or fewer barriers to care [31,32]. Our analysis suggests that mobile individuals in this urban Ugandan community are more likely to be employed and have higher incomes, which may indicate higher socioeconomic status and thus lower risk for disease and better access to care. Other mobile populations have been shown to be at increased risk for TB, and interventions targeting these populations have been successful in, for example, providing TB diagnostic and treatment services for truck drivers in India [33] and nomadic populations in Nigeria [13] and Iran [12]. Migrant-centered care, including mobile clinics, expanding service hours, flexible treatment options, or health passports may be appropriate services for such populations [3].

Limitations of our study include the incomplete measurement of mobility. Mobility questions were assessed via self-report in patient interviews and may be subject to misclassification or bias. Additionally, we asked a short set of questions that may not capture every component of mobility nor the impetus for mobility (marriage, work, education, recreation travel, etc.). GPS trackers have been used in other studies and can capture continuous information that can be used to calculate additional indicators [34,35] and may provide more reliable information [34,36]. However, our approach using an LCA contributes to the development of a mobility definition that may be applied to other populations in which such technology is unavailable. Our interviews were conducted after TB diagnostic evaluations; participants, particularly those who were diagnosed with TB, may therefore have been symptomatic for substantial periods of time. A lack of mobility may thus reflect the effects of TB disease itself (i.e., be subject to reverse causality), which could counterbalance any increased TB risk that might be associated with increased mobility; a similar relationship has been suggested for HIV [37]. Prospective data collection may help clarify these causal relationships between mobility and risk of disease. Additionally, our population of urban individuals either seeking TB diagnostic services at health facilities or participating in community-based TB testing may not represent the general population. Cases and controls were matched by place and time of diagnosis and in order to increase comparability to each other, but further studies of the association between mobility and TB risk in other populations are therefore warranted.

## Conclusions

We developed a data-driven, straightforward measure of mobility (traveling >3km from residence ≥2 times per month) among patients seeking care at health facilities in a densely populated community of Kampala, Uganda. Mobility was associated with decreased risk of TB

disease; this was counter to our original hypothesis and may be in part due to the higher socio-economic status of mobile individuals in this population. Additional research to better measure and classify mobility, including associations between mobility and infectious disease risk, should consider direct measurement of movement, prospective data collection, and inquiries into the reason for travel. These data should be evaluated in a diverse array of populations in order to deepen our understanding of the complex construct of human mobility and the degree to which mobility contributes to the spread of TB and other infectious diseases.

## Supporting information

**S1 Fig. Distribution of frequency and duration of travel >3 km.**
(PNG)

**S1 Table. Model selection.**
(DOCX)

**S2 Table. Estimated marginal means for latent classes of mobility.**
(DOCX)

**S3 Table. Estimated marginal means for latent classes of mobility stratified by case status.**
(DOCX)

**S4 Table. Estimated marginal means for latent classes of mobility stratified by enrollment method.**
(DOCX)

**S5 Table. Estimated marginal means for latent classes of mobility—sensitivity analysis using Poisson-distributed variables.**
(DOCX)

**S6 Table. Estimated marginal means for latent classes of mobility—sensitivity analysis for dichotomizing continous variables.**
(DOCX)

**S7 Table. Mobility and duration of TB related symptoms (prior to enrollment) among symptomatic individuals with TB who reported symptoms.**
(DOCX)

**S1 Appendix. Case and control interview tool: movement and mobility section.**
(PDF)

## Acknowledgments

We thank the STOMP-TB field team for their efforts in case finding and data collection. We also thank Kampala City Council Authority, the Uganda National TB and Leprosy Control Programme, and the staff and patients at Kisugu Health Center, Alive Medical Services, International Hospital Kampala (Touch-Namuwongo), and Meeting Point Clinic for their participation, as well as the community members in our study area for their participation and support.

## Author Contributions

**Conceptualization:** Katherine O. Robsky, David W. Dowdy.

**Data curation:** Katherine O. Robsky, David Isooba, Olga Nakasolya, James Mukiibi, Annet Nalutaaya, Peter J. Kitonsa.

**Formal analysis:** Katherine O. Robsky, Annet Nalutaaya, Emily A. Kendall, David W. Dowdy.

**Funding acquisition:** Katherine O. Robsky, Emily A. Kendall, Achilles Katamba.

**Investigation:** David Isooba, Olga Nakasolya, James Mukiibi, Annet Nalutaaya, Peter J. Kitonsa, Caleb Kamoga, Achilles Katamba.

**Methodology:** Katherine O. Robsky, Yeonsoo Baik, Emily A. Kendall, David W. Dowdy.

**Project administration:** Peter J. Kitonsa, Caleb Kamoga.

**Supervision:** Achilles Katamba, David W. Dowdy.

**Writing – original draft:** Katherine O. Robsky, Emily A. Kendall, Achilles Katamba, David W. Dowdy.

**Writing – review & editing:** Katherine O. Robsky, David Isooba, Olga Nakasolya, James Mukiibi, Annet Nalutaaya, Peter J. Kitonsa, Caleb Kamoga, Yeonsoo Baik, Emily A. Kendall, Achilles Katamba, David W. Dowdy.

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
