## [Decision Letter · Decision Letter 0]

8 Apr 2021

PONE-D-21-03202

Characterization of geographic mobility among participants in facility- and community-based tuberculosis case finding in urban Uganda

PLOS ONE

Dear Dr. Robsky,

Thank you for submitting your manuscript to PLOS ONE. After careful consideration, we feel that it has merit but does not fully meet PLOS ONE’s publication criteria as it currently stands. Therefore, we invite you to submit a revised version of the manuscript that addresses the points raised during the review process.

We look forward to receiving your revised manuscript.

Kind regards,

Limakatso Lebina, MBChB

Academic Editor

PLOS ONE

Journal Requirements:

In the methods section please clarify the tool used to documents to collect self reported information on mobility. If this is a questionnaire/survey it is not under a copyright more restrictive than CC-BY, please include a copy, in both the original language and English, as Supporting Information.

Please provide additional details regarding participant consent. In the ethics statement in the Methods and online submission information, please ensure that you have specified whether consent was written or verbal/oral. If consent was verbal/oral, please specify: 1) whether the ethics committee approved the verbal/oral consent procedure, 2) why written consent could not be obtained, and 3) how verbal/oral consent was recorded.”

Please include captions for your Supporting Information files at the end of your manuscript, and update any in-text citations to match accordingly. Please see our Supporting Information guidelines for more information: http://journals.plos.org/plosone/s/supporting-information.

Additional Editor Comments:

Thank you for submitting an interesting manuscript.

There are a few issues that need to be clarified in the methods sections on how data was collected and analysis done. 50 000 people in a 2.2KM radius sounds very crowded. More information on the setting would help one understand why 3km was considered mobility. What could people access within the 3km radius. How did you explain to the participants the more than 3km radius travel?

Reviewers' comments:

Reviewer's Responses to Questions

**Comments to the Author**

1. Is the manuscript technically sound, and do the data support the conclusions?

Reviewer #1: Yes

Reviewer #2: Yes

2. Has the statistical analysis been performed appropriately and rigorously? 

Reviewer #1: Yes

Reviewer #2: Yes

3. Have the authors made all data underlying the findings in their manuscript fully available?

Reviewer #1: Yes

Reviewer #2: No

4. Is the manuscript presented in an intelligible fashion and written in standard English?

Reviewer #1: Yes

Reviewer #2: Yes

5. Review Comments to the Author

Reviewer #1: Characterization of geographic mobility among participants in facility- and community based

tuberculosis case finding in urban Uganda

Summary

Recognising that international and internal migration are risk factors for TB, the authors set to define geographic mobility in two Uganda urban population including, individuals attending for care at health facilities and those identified through community-based TB case finding. Through interviews with participants (individuals diagnosed with TB [cases] and individuals in the same population group that were not diagnosed with TB [controls] data was collected on specific characteristics/components of mobility defined 1 priori. Latent class analysis was done using data on the eight components collected from all 699 participants enrolled in the study. characteristics were used to identify and define mobility. Participants were assigned to the most probable mobility class. The authors then went further to determine the association of mobility and several factors including, demographic, socioeconomic and TB risk factors. They found that mobility was associated with decreased risk of TB.

Minor comments:

1. Understanding internal migration and how it may influence TB transmission in general communities is important and the authors present an interesting analysis. The results suggest that mobility may be associated with a reduction in TB risk, which is counterintuitive.

a) Given the information on the indicators was collected through self-report, is it possible that the way the questions were asked/responded to might and the type of questions asked would have influenced the outcome?

b) Also just wondering what whether in this setting the prevalence of TB was homogenous or there are specific hotspots for transmission. Is there information on most places people travelled to and whether these were hotspots for transmission.

c) Did the team look at commonly used mode of transport? Could it have been a useful indicator to measure given the likely risk of using overcrowded public transport might increase the risk of getting TB?

2. The authors acknowledged the STOMP-TB study team in the manuscript. Was this study part of or a sub-study of the STOMP-TB study or completely unrelated? If it is it might be good to give some information on the parent study.

3. The data presented suggested that there is need to conduct more research to improve assessment of mobility and how it affects risk of TB. What would the authors suggest as important data to collect in order to refine the definition of mobility? It will be good to make this more explicit in the discussion?

Reviewer #2: Characterization of geographic mobility among participants in facility- and community- based tuberculosis case finding in urban Uganda

The authors sought to characterize mobility among TB diagnosed individuals and the association of mobility with TB risk. They conducted a case-control study where they defined a case as an individual diagnosed at facility or through community based case-finding, and these were compared to 1 or 2 negative controls depending on place of diagnosis

LCA was used to characterize mobility using self-report on 8 mobility items which were dichotomized. The latent classes for mobility were mainly defined by travel >3km >2 times per month and this item was ultimately the main exposure variable used in the mobility-TB risk analysis

Comments

1. Line 87 suggests that matching was done on facility and approximate date of enrollment and residence. This makes one wonder about age and possibly other socio-demographic characteristics such as sex – understandably, there would be more males with TB than females. One would think that at least age and employment status should be considered for matching in addition to where and when an individual was diagnosed.

2. Furthermore, can the authors please explain the significance of matching by place and time of diagnosis? I think this is worth having in the discussion section.

3. There is loss of information in the data when you dichotomize the items used in defining mobility although this has an advantage that the model places equal weight on all the items. Ultimately, the number of trips >3km was used as a proxy for mobility, but it would be interesting to conduct an exploratory analysis (such as a penalized regression or classification regression approach) that includes each item without dichotomizing the variable.

4. Line 145 – would be good to define what the potential confounders are e.g. socio-demographic/economic characteristics?

5. Line 204 paragraph has some double negatives that make the message unclear

6. PLOS authors have the option to publish the peer review history of their article (what does this mean?). If published, this will include your full peer review and any attached files.

Reviewer #1: No

Reviewer #2: No

---

## [Author Response · Author response to Decision Letter 0]

27 Apr 2021

PONE-D-21-03202

Dear Dr. Lebina,

Thank you for your consideration of our manuscript. We have addressed the reviewers’ concerns in the revised manuscript and have detailed these responses to specific questions and concerns below. We believe that these revisions have strengthened the manuscript and are appreciative to the two reviewers for these very helpful comments. We look forward to hearing from you when an editorial decision has been made. 

Sincerely,

Katherine Robsky 

Journal Requirements:

 Authors’ response: Thank you for reminding us of the style requirements. We have made modifications to the figure file names and supporting information to align with the requirements.

2. In the methods section please clarify the tool used to documents to collect self reported information on mobility. If this is a questionnaire/survey it is not under a copyright more restrictive than CC-BY, please include a copy, in both the original language and English, as Supporting Information.

Authors’ response: We have added clarifying language to the methods section under “Measurement of components of geographic mobility” (lines 101-103): We defined a priori eight components of mobility (Table 1) and collected self-reported information through participant interviews using a tool developed for and pilot tested in this study population (Appendix S1).” We have also uploaded the interview tool as Supporting Information.

3. Please provide additional details regarding participant consent. In the ethics statement in the Methods and online submission information, please ensure that you have specified whether consent was written or verbal/oral. If consent was verbal/oral, please specify: 1) whether the ethics committee approved the verbal/oral consent procedure, 2) why written consent could not be obtained, and 3) how verbal/oral consent was recorded.”

Authors’ response: Thank you for pointing out this omission. We have clarified that written consent was provided in the “Ethical considerations” section (lines 154-155): “All participants provided written informed consent (or written assent and parental consent for those 15-17 years old) for all study activities.”

Authors’ response: Thank you for pointing out this omission. We have added the list of supporting files to the manuscript (starting line 429), with captions, and updated the in-text citations accordingly.

Authors’ response: We have reviewed the reference list and have updated it to ensure its accuracy.

Additional Editor Comments:

Thank you for submitting an interesting manuscript.

There are a few issues that need to be clarified in the methods sections on how data was collected and analysis done. 50 000 people in a 2.2KM radius sounds very crowded. More information on the setting would help one understand why 3km was considered mobility. What could people access within the 3km radius. How did you explain to the participants the more than 3km radius travel?

Authors’ response: Thank you for this comment. We have added a description of “densely populated” to our description of the study area to make the setting more clear to the reader (line 74). We have also added additional detail in the Measurements of components of geographic mobility (lines 109-113) describing the use of 3km as a cutoff: “We defined travel outside of the participant’s neighborhood as travel >3 km from their primary residence; this represents leaving the study area; likely use of a car, bus, or taxi; and the potential for TB exposure outside the immediate vicinity of the participant’s local community. Research assistants provided landmarks and Google maps to help participants identify locations they may have visited >3 km from home.”

Comments to the Author

5. Review Comments to the Author

Reviewer #1: Characterization of geographic mobility among participants in facility- and community based

tuberculosis case finding in urban Uganda

Summary

Recognising that international and internal migration are risk factors for TB, the authors set to define geographic mobility in two Uganda urban population including, individuals attending for care at health facilities and those identified through community-based TB case finding. Through interviews with participants (individuals diagnosed with TB [cases] and individuals in the same population group that were not diagnosed with TB [controls] data was collected on specific characteristics/components of mobility defined 1 priori. Latent class analysis was done using data on the eight components collected from all 699 participants enrolled in the study. characteristics were used to identify and define mobility. Participants were assigned to the most probable mobility class. The authors then went further to determine the association of mobility and several factors including, demographic, socioeconomic and TB risk factors. They found that mobility was associated with decreased risk of TB.

Minor comments:

1. Understanding internal migration and how it may influence TB transmission in general communities is important and the authors present an interesting analysis. The results suggest that mobility may be associated with a reduction in TB risk, which is counterintuitive.

a) Given the information on the indicators was collected through self-report, is it possible that the way the questions were asked/responded to might and the type of questions asked would have influenced the outcome?

Authors’ response: We agree that there may be limitations in how the questions were asked that could cause bias in this analysis. As we mention in the discussion section (line 290-295), our interviews were conducted after participants had been tested for TB. Patients with TB symptoms may have had those symptoms for a long period of time, and thus their observed lower mobility may reflect effects of their illness; it is therefore possible that the lower mobility among TB cases is in fact due to their disease, rather than mobility itself being protective of disease. However, our inclusion of community-diagnosed cases, who reported a shorter median duration of symptoms (4 weeks vs. 8 weeks in facility-diagnosed cases, lines 233-237), but still less likely to be mobile compared to TB-negative controls, makes this possibility less likely.

As is the nature of case-control studies, it is possible that there was differential recall between cases and controls. However, in that case one would expect cases be more likely to remember and report an exposure (because they are thinking about potential causes of their newly diagnosed illness); as we observed cases were less likely to report mobility, we do not believe this is what caused our somewhat counterintuitive finding. Instead, as we mention in the discussion (lines 256-257, and 284-285) we believe that mobility, in at least some instances, may represent increased socioeconomic status, as the travel out beyond 3 km may be due to working in a higher-income area or the ability to travel itself may indicate a certain socioeconomic status (if one owns a car, for example.)

b) Also just wondering what whether in this setting the prevalence of TB was homogenous or there are specific hotspots for transmission. Is there information on most places people travelled to and whether these were hotspots for transmission.

Authors’ response: Previous analyses of this study area have demonstrated that there is substantial heterogeneity of TB based on home residence even within the small study area[1]. We controlled for this by matching on zone of residence (for community-enrolled participants, line 94-97) or whether the participant lived within or outside the study area (for facility-enrolled participants, line 87-89). However, we unfortunately do not have information on where people traveled to, so we are unable to determine if these are likely hotspots where individuals could have been exposed. In response to this comment and the Editor’s suggestion, we have included our questionnaire as an Appendix to make it clear to readers what data were – and were not – elicited.

Reference: 

1. Robsky KO, Kitonsa PJ, Mukiibi J, Nakasolya O, Isooba D, Nalutaaya A, et al. Spatial distribution of people diagnosed with tuberculosis through routine and active case finding: a community-based study in Kampala, Uganda. Infect Dis Poverty. 2020;9: 73. doi:10.1186/s40249-020-00687-2

c) Did the team look at commonly used mode of transport? Could it have been a useful indicator to measure given the likely risk of using overcrowded public transport might increase the risk of getting TB?

Authors’ response: We agree that crowded public transport may put people at increased risk of TB exposure, which was our rationale for including frequency of visiting a taxi park as a measure of mobility. We have added clarification describing Ugandan taxis to Table 1 (lines 124-126): “Note that in Uganda, a taxi refers to a van or minibus that picks up and drops off multiple passengers along a specific route, akin to a public bus route in other settings.” Because we did not have a measure of how often a taxi was used (or any information regarding the distance traveled, how crowded the taxi was, or if windows were open), we used frequency of visiting a taxi park as a proxy measure for this exposure. This was included in the latent class analysis, which showed that the mobile class was approximately twice as likely to report visiting a taxi stage than the non-mobile class (36% vs. 18%, Figure 1),

2. The authors acknowledged the STOMP-TB study team in the manuscript. Was this study part of or a sub-study of the STOMP-TB study or completely unrelated? If it is it might be good to give some information on the parent study.

Authors’ response: Thank you for this comment. We have clarified that this was conducted as part of the STOMP-TB study in line 75-79, with a reference to a manuscript that describes the study in greater detail: “This analysis was conducted as part of the STOMP-TB (Strategies for Treating, Observing, Managing, and Preventing Tuberculosis) study, an ongoing population-based study of TB transmission in a densely populated area consisting of 37 contiguous administrative zones in Kampala, Uganda (estimated population: 50,436, total area 2.2 km2 [28]). The STOMP-TB study enrolls patients through two mechanisms: health facility TB diagnostic testing (from May 2018-July 2021) and active community outreach and testing (Phase 1 conducted from February to December 2019).” 

We have further clarified our subset of the data beginning on line 80: “For this analysis, we enrolled all consenting patients age ≥15 years who presented for TB testing at one public and three private outpatient TB Diagnosis and Treatment Units in the study area from February 2019 (when we began collecting information on mobility) to August 2020 (the administrative stopping date for this analysis).”

3. The data presented suggested that there is need to conduct more research to improve assessment of mobility and how it affects risk of TB. What would the authors suggest as important data to collect in order to refine the definition of mobility? It will be good to make this more explicit in the discussion?

Authors’ response: We discuss in our limitations section (lines 295-311) that additional questions that capture mobility, direct measurement of mobility such as using GPS trackers, prospective data collection, and measurement of mobility in non-urban settings may help refine the definition of mobility. We have added clarification that additional questions to capture mobility should also address the reason for mobility, such as due to marriage, work, education, or recreational travel (lines 298-299), which may affect the individual’s TB exposure and access to care.

We have further emphasized these suggestions in our conclusion (line 319-325): “Additional research to better measure and classify mobility, including associations between mobility and infectious disease risk, should consider direct measurement of movement, prospective data collection, and inquiries into the reason for travel. These data should be evaluated in a diverse array of populations in order to deepen our understanding of the complex construct of human mobility and the degree to which mobility contributes to the spread of TB and other infectious diseases.”

Reviewer #2: Characterization of geographic mobility among participants in facility- and community- based tuberculosis case finding in urban Uganda

The authors sought to characterize mobility among TB diagnosed individuals and the association of mobility with TB risk. They conducted a case-control study where they defined a case as an individual diagnosed at facility or through community based case-finding, and these were compared to 1 or 2 negative controls depending on place of diagnosis

LCA was used to characterize mobility using self-report on 8 mobility items which were dichotomized. The latent classes for mobility were mainly defined by travel >3km >2 times per month and this item was ultimately the main exposure variable used in the mobility-TB risk analysis

Comments

1. Line 87 suggests that matching was done on facility and approximate date of enrollment and residence. This makes one wonder about age and possibly other socio-demographic characteristics such as sex – understandably, there would be more males with TB than females. One would think that at least age and employment status should be considered for matching in addition to where and when an individual was diagnosed.

Authors’ response: Thank you for this comment – it is always challenging to balance the right number of factors for matching. In this study, we did not match on employment status because this was not readily available from the facility treatment registers or our brief screening form (data which we collected from >12,000 individuals and thus had to be brief and non-intrusive). Thus, matching on employment status would have resulted in a non-representative sample. For age and sex, we did not match on these characteristics because matching would have removed any association between these variables and TB status – an association that (unlike facility or date of enrollment) we were interested to observe and report in this community. While such matching might have been useful for this specific analysis, it therefore would potentially have been detrimental to the larger study.

2. Furthermore, can the authors please explain the significance of matching by place and time of diagnosis? I think this is worth having in the discussion section.

Authors’ response: The intention of matching by place and time of location was to reduce the difference health care access and care-seeking behaviors between cases and controls, in order to improve our ability to compare these two groups. Health facility controls represent individuals who would have been diagnosed at the health facility if did have TB, while community controls represent individuals who would be detected by community-based TB testing. We have added further clarification to that while this study design increases the validity of our comparisons between the groups, it does mean that they are less representative of a random sample in the population (lines 313-314): “Additionally, our population of urban individuals either seeking TB diagnostic services at health facilities or participating in community-based TB testing may not represent the general population. Cases and controls were matched by place and time of diagnosis and in order to increase comparability to each other, but further studies of the association between mobility and TB risk in other populations are therefore warranted.”

3. There is loss of information in the data when you dichotomize the items used in defining mobility although this has an advantage that the model places equal weight on all the items. Ultimately, the number of trips >3km was used as a proxy for mobility, but it would be interesting to conduct an exploratory analysis (such as a penalized regression or classification regression approach) that includes each item without dichotomizing the variable.

Authors’ response: We agree that dichotomizing continuous variables can lead to the loss of information. We did consider different cutoffs as sensitivity analyses (Tables S8), but based on this suggestion have further constructed LCA models where the number of trips >3 km and the duration of those trips are individually included as Poisson-distributed variables rather than binary variables (Table S7). We have clarified the sensitivity analyses in the methods (line 114-118): “We used the overall median in the population to define the cutoff for frequency and duration of travel >3 km from home; we also considered the 75th quartile as a cutoff and modeling each of the variables as Poisson-distributed (non-dichotomized) as sensitivity analyses. The remaining measures and their categorization are described in Table 1.” Additional clarifications were made in Table 1 (Components of Geographic Mobility) and have also added a histogram for these two measures (Figure S2) indicating the cut-off for dichotomization. 

In the LCA modeling frequency of travel as a Poisson-distributed variable, the mean number of trips in the non-mobile class was 1.5, which suggests that using a cutoff of 2 for simple dichotomization was reasonable. These analyses demonstrate that the decision of whether to dichotomize this variable versus treat it as continuous variable do not substantively affect our conclusions. 

We considered penalized regression and classification/regression trees in response to this comment, but these methods typically require an observed, rather than a latent, outcome. The number of trips >3km was used as a proxy for mobility only after observing that this variable had perfect overlap with the calculated latent construct of mobility – we did not define this as a proxy for mobility on an a-priori basis.

4. Line 145 – would be good to define what the potential confounders are e.g. socio-demographic/economic characteristics?

Authors’ response: Thank you for pointing out this omission. We have added further clarification to the process of identifying and adjusting for confounders (lines 149-152): “All potential confounders were identified a priori as characteristics believed to be associated with mobility and previously shown to be associated with TB disease, including demographic, socioeconomic, and clinical and behavioral risk factors, and were included in the final model regardless of statistical significance.” We also include our survey instrument as an appendix, in response to this and other comments made by the Editor and Reviewer 1.

5. Line 204 paragraph has some double negatives that make the message unclear

Authors’ response: We have reworded the final sentence to remove the double negative: “Unlike among cases, there was no association between mobility and degree completion among controls (PR 0.91, 95% CI 0.54, 1.52). “

---

## [Editor Report · Decision Letter 1]

4 May 2021

Characterization of geographic mobility among participants in facility- and community-based tuberculosis case finding in urban Uganda

PONE-D-21-03202R1

Dear Dr. Katherine O Robsky,

We’re pleased to inform you that your manuscript has been judged scientifically suitable for publication and will be formally accepted for publication once it meets all outstanding technical requirements.

Kind regards,

Limakatso Lebina, MBChB

Academic Editor

PLOS ONE

Additional Editor Comments (optional):

Thank you for responding and addressing all the comments.
---

## [Editor Report · Acceptance letter]

6 May 2021

PONE-D-21-03202R1 

Characterization of geographic mobility among participants in facility- and community-based tuberculosis case finding in urban Uganda 

Dear Dr. Robsky:

I'm pleased to inform you that your manuscript has been deemed suitable for publication in PLOS ONE. Congratulations! Your manuscript is now with our production department. 

Kind regards, 

on behalf of

Dr. Limakatso Lebina 

Academic Editor

PLOS ONE